# New Approaches for Pb(II) Removal from Aqueous Media Using Nanopowder Sodium Titanosilicate: Kinetics Study and Thermodynamic Behavior

**DOI:** 10.3390/ijms241813789

**Published:** 2023-09-07

**Authors:** Ionela Carazeanu Popovici, Simona Dobrinaș, Alina Soceanu, Viorica Popescu, Gabriel Prodan, Ichinur Omer

**Affiliations:** 1Chemistry and Chemical Engineering Department, Ovidius University of Constanta, 900527 Constanta, Romania; icarazeanu@univ-ovidius.ro (I.C.P.); sdobrinas@univ-ovidius.ro (S.D.); vpopescu@univ-ovidius.ro (V.P.); 2Electron Microscopy Laboratory, Department of Physics, Ovidius University of Constanta, 900527 Constanta, Romania; gprodan@univ-ovidius.ro; 3Civil Engineering Faculty, Ovidius University of Constanta, 900527 Constanta, Romania; ichinur.omer@univ-ovidius.ro

**Keywords:** sodium titanosilicate, lead removal, adsorption, kinetics and isotherm models

## Abstract

Microporous sodium titanosilicate, Na_2_TiSiO_5_, has been successfully prepared using the sol–gel method. The structural and morphological characterization of synthesized product has been made via thermal analyses (TG-DTG), X-ray diffraction (XRD), and electron microscopy (SEM and TEM). Adsorption properties of the synthesized Na_2_TiSiO_5_ nanopowder for Pb(II) removal of aqueous media was investigated in different experimental conditions such as the contact time, the initial metal concentration, the pH, and the temperature. The Pb(II) adsorption on Na_2_TiSiO_5_ was discussed according to the kinetics and thermodynamics models. The adsorption kinetics of Pb(II) have been better described by the PS-order kinetic model which has the highest fitting correlation coefficients (R^2^: 0.996–0.999) out of all the other models. The adsorption results have been successfully fitted with the Langmuir and Redlich–Paterson models (R^2^: 0.9936–0.9996). The calculated thermodynamic parameters indicate that the Pb(II) adsorption is an endothermic process, with increased entropy, having a spontaneous reaction. The results have revealed a maximum adsorption capacity of 155.71 mg/g at 298 K and a very high adsorption rate at the beginning, more than 85% of the total amount of Pb(II) being removed within the first 120 min, depending on the initial concentration.

## 1. Introduction

The high level of toxic heavy metals we have been confronted with more and more in the recent years is the consequence of industrialization and urbanization of modern societies. This can be easily observed in almost every industrial activity involving leakage and redistribution of heavy metals, such as mining, metallurgy, steel and iron, electrolysis, electroosmosis, electroplating, and leatherworking. On the other hand, all these industrial activities have led to the exhaustion of metal land mineral resources. The wastewaters produced from these industrial activities represents serious environmental problems and a threat for human health and the natural ecosystems [1,2,3,4,5].

The wastewaters are among the most hazardous factors in the chemical-intensive industries since these procedures involve discharging of a large amount of contaminated metals such as Pb, As, Cr, Ni, Cu, Cd, Zn, and Hg [6,7,8]. The most dangerous effects on human health and natural ecosystems are brought by the long-term exposure to those solvated metal ions. Pb represents one of the common contaminants of industrial wastewaters. A daily intake of lead can cause various and serious disorders to human organism, such as damages to liver and kidney as well as reduction in hemoglobin formation, encephalopathy or mental retardation, infertility, and abnormalities in pregnant women [9,10,11,12,13,14,15,16]. Therefore, the most important issue to be studied on this matter is the minimization or complete removal of the heavy metals from the aqueous media. This can be carried out by transferring the heavy metal ions from the aqueous media to the solid phase via the adsorption process [17]. A strong affinity between the solid phase and the targeted metal ions is required in order to bind them irreversibly under ambient conditions; at the same time, it is important to have the ability to release them under different conditions, so that it can be regenerated for further use [18,19,20,21,22]. Some different solid phases have been found useful in the removal of heavy metals from aqueous media, such as metal oxides, activated carbons, clay minerals, biosorbents, and zeolites [7,21,23,24,25,26,27,28,29,30,31,32,33,34,35,36,37,38,39,40,41,42,43,44,45,46,47,48,49,50]. Successful steps have been made in the recent years towards finding new materials capable of improving the separation process such as some microporous titanosilicates with potential applications in catalysis, separation processes, or ion exchange [17]. To optimize the structure of these new materials, new synthesis methods should be used. On the other hand, it is very important to have a comprehensive understanding of chemical and morphological characterization of them, in order to better understand their properties [18].

Titanosilicates are constructed from interconnected polyhedra, octahedra or pentahedra, with Ti as the central atom and SiO_4_ tetrahedra. The negative charge on the Ti–O groups is compensated by cations that can be exchanged. Variations in the incorporation of these structural components leads to the formation of framework and layered or dense structures [14,15,16]. 

Oleksiienko et al. [17] consider that titanosilicates belong to the AM-*n* family, the newest group of sorption materials. AM-1 is the first member of the group and sodium titanosilicate, Na_2_TiSiO_5_ is one of the compounds included in the group. It is known that the Na_2_TiSiO_5_ has three polymorphic modifications, two tetragonal and one orthorhombic [18,19]. Titanosilicates materials with their framework and layered structures have the highest adsorption potential and a high ion-exchange rate. They also have a high sorption capacity and selectivity at a broad pH range, in addition to remaining ‘‘uncontaminated” by other cations. Titanosilicates are the new catalysts or molecular sieves and ion exchangers. The efforts of many researchers are directed towards explication of the crystal growth mechanism with the purpose to extend synthesis methods in order to obtain titanosilicate materials with desired properties [17]. Some recent researches aimed to modify the known titanosilicates and obtain new titanosilicates as adsorbents capable to remove heavy metals, organic pollutants, and radioactive pollutants from wastewater [16,17].

The specific property of the crystalline structure of Na_2_TiSiO_5_ consists of the existence of a rare five coordinated titanium. The crystalline structure of Na_2_TiSiO_5_ is made up of layers of TiO_5_ square pyramid and SiO_4_ tetrahedra joined at corners and separated by layers of sodium ions [20]. This structure assures a good ion-exchange property in Na_2_TiSiO_5_; thus, it can be investigated as a new adsorbent [21]. 

Major research about the synthesis and characterization of microporous materials such as vanadosilicates, zirconosilicates, and titanosilicates has been reported by Rocha et al. [22,23,24]. Mesoporous materials with ordered pore structure and large surface area have been proved to be very efficient with such applications, ranging from air to water purification. Modified or untreated mesoporous silica has been successfully applied to different adsorption pollutants too [38,39]. Such kinds of materials have considerable potential in adsorption applications because of their structure with uniform pore distribution, large specific surface area, and large pore volume.

The aim of this work Is to synthesize sodium titanosilicate, Na_2_TiSiO_5_, and investigate its application for the removal of the Pb(II) from aqueous solutions. The new adsorbent, Na_2_TiSiO_5_, has been prepared via the sol–gel method. The structural and morphological characterization of Na_2_TiSiO_5_ has been made using thermal analyses (TG-DTG), X-ray diffraction (XRD), and electron microscopy (SEM and TEM) analysis. 

Adsorption properties of the synthesized Na_2_TiSiO_5_ nanopowder have been investigated in different experimental conditions, such as contact time, initial metal concentration, pH, and temperature. 

The Pb(II) adsorption on Na_2_TiSiO_5_ has been discussed in accordance to the kinetics, and thermodynamics models. Five kinetic models, including the PF-order equation, the PS-order equation, the intraparticle diffusion equation, the Elovich equation and the Bangham’s equation have been selected to follow the adsorption process of the Pb(II) ions on Na_2_TiSiO_5_. The Pb(II) adsorption isotherms were also modeled using Langmuir, Freundlich, Halsey, Temkin, Redlich–Paterson, and Dubinin–Kaganer–Radushkevich isotherm models.

## 2. Results and Discussion

### 2.1. Characterization of Mesoporous Sodium Titanosilicate, Na_2_TiSiO_5_

#### 2.1.1. Thermal Studies

TG–DTG curves display the formation temperature of the Na_2_TiSiO_5_ grown using the modified sol–gel method (Figure 1).

Thermal analysis (TG–DTG) performed in the temperature range 20–1000 °C showed a total mass loss of 20.11%. According to the TG curve the most significant mass loss take place in the temperature range from 20 to 300 °C that may be assigned to the loss of water molecules [17,25]. This interpretation of results is supported by the presence of two endothermic peaks on the DTG curve; one peak at 113 °C attributed to water adsorbed in the pores (physically adsorbed water) and the other one at 278 °C attributed to desorption of structural water (hydrated water removal) [44]. After that, a continuous slight mass loss is observed up to 900 °C. This mass loss can be attributed to the decomposition reaction of precursor and formation of Na_2_TiSiO_5_.

#### 2.1.2. X-ray Diffraction

The powder XRD phase analysis emphasizes that bulk quantities of Na_2_TiSiO_5_ particles can be obtained using the sol–gel method at low temperatures (below 500 °C), as confirmed by TG–DTG. Figure 2 shows the XRD pattern of synthesized Na_2_TiSiO_5_ after thermal treatment at 800 °C, for 2 h.

The XRD spectra permitted the röntgenographic deceleration of the Na_2_TiSiO_5_ by its specific interference at 2.86 Å, 3.30 Å, and 1.77 Å and indexing of additional low-angle reflections—110, 101, 200, 002, 211, 220, and 301 (JCPDS 01-086-1615) [26,27,28,29]. Crystallization of the Na_2_TiSiO_5_ precursor began at 500 °C and was complete at 800 °C.

#### 2.1.3. Fourier Transform Infrared Spectroscopy (FTIR)

The FTIR spectrum recorded in the range of 400–2000 cm^−1^ (Figure 3) for the Na_2_TiSiO_5_ nanopowder is in correlation with literature data [17,30]. The major absorption bands correspond to the asymmetric and symmetric stretching vibrations (at 850.6 cm^−1^ assigned to the asymmetric stretching of Ti–O–Ti bridges, at 912.2 cm^−1^ assigned to the asymmetric stretching of Si–O–Ti bridges and at 576.2 cm^−1^ assigned to the asymmetric stretching of Si–O–Si bridges [17,19,30,31]. 

The sharp band at 1633 cm^−1^ can be attributed to the interstitial water molecules and HO–H bending. Strong and broad peaks in the range 450–580 cm^−1^ and 800–1100 cm^−1^ appeared in the spectra of synthesized Na_2_TiSiO_5_ powder can be assigned to the presence of silicate groups [31,32].

#### 2.1.4. Electron Microscopy

Surface morphology, texture, and particles size of Na_2_TiSiO_5_ nanopowder have been investigated by SEM and TEM [25,32]. The Figure 4 presents the SEM micrograph with EDX spectra insert for of the synthesized Na_2_TiSiO_5_. The EDX analysis qualitatively confirmed the purity of the Na_2_TiSiO_5_ powder.

The morphology and mean diameter have been studied using a bright-field (BF) TEM micrograph (Figure 5a). In the HRTEM image of the synthetized Na_2_TiSiO_5_ are inserted the histogram of Feret diameter. The particle’s diameters have been evaluated using the mean value of distances between the pairs of parallel tangents to the projected outline of the particle (Feret’s diameter). The mean diameter has been calculated assuming a lognormal distribution of experimental data [25,32].

The HRTEM image of the sample confirm that the synthetized Na_2_TiSiO_5_ has a mesoporous structure. The HRTEM image reveals Na_2_TiSiO_5_ nanoparticles in the size range of 3–16 nm and a mean diameter of 7.45 nm.

### 2.2. Effect of Phase Contact Time, pH and Adsorption Kinetic Studies

The adsorption’s kinetics process describes the degree of metal ions removal, and it is one of the most important characteristics of the adsorption process efficiency. Generally, the sorption of metal ions increases with the increase of the contact time. The sorption reactions take place with an important rate in the initial phases and are gradually slowed down until the equilibrium state is reached [8,13,18].

The physical–chemical parameters that influence the adsorption capacity and retention efficiency of PB(II) ions on Na_2_TiSiO_5_ are contact time, pH, initial concentration of Pb(II) ions and temperature. Figure 6 presents the influence of the contact time and the initial ion concentration of Pb(II) on the adsorption capacity and retention efficiency of Pb(II) ions at different temperatures.

In the first stage, up to a contact time of 60 min, both the adsorption capacity and the retention efficiency of Na_2_TiSiO_5_ increase rapidly due to the fast adsorption of Pb(II). This first stage is attributed to the diffusion stage, when the most available adsorbent layers on the Na_2_TiSiO_5_ surface are occupied very quickly. In the second stage, up to 120 min, Pb(II) adsorption increases gradually in time until equilibrium is reached. This second transition phase is attributed either to external diffusion through the layer at the adsorbent–solution interface, or to internal diffusion in the pores of the adsorbent particle. The third stage represents the equilibrium stage where the removal of the adsorbent material becomes almost insignificant, because of the depletion of the active adsorption sites. Upon reaching equilibrium, more than 85% of the initial amount of Pb(II) was adsorbed on Na_2_TiSiO_5_. Increasing the initial Pb(II) ion concentration from 1 × 10^−2^ M to 7 × 10^−2^ M leads to more unabsorbed Pb(II) ions in the solution due to the saturation of the active adsorption sites, leading to a low removal efficiency.

The temperature also influences the adsorption capacity and the removal efficiency of Pb(II). Increasing the temperature from 293 K to 313 K leads to a slight intensification of the Pb(II) removal process.

This results show that the contact time required to obtain a maximum Pb(II) adsorption capacity is 120 min. The maximum contact time required for the removal of Pb(II) is comparable to some results mentioned in some of the earlier papers regarding the adsorption of Pb(II) ions on different adsorbents, which reports equilibrium times of 50–120 min [8,33,36,38].

Huang [48] asserts that the initial pH significantly influences the adsorption processes of metal ions; it influences both the states of the functional groups on the surface of the adsorbent, and the type of the metal ions in solution. It is well known that in the solution of pH 2.0–8.0, there are three forms of lead species: Pb^2+^, Pb(OH)^+^, and Pb(OH)_2_ [6,18,33,38,48]. The Pb(II) adsorption amount from sodium titanosilicate, Na_2_TiSiO_5_, versus pH values is illustrated in the Figure 7. 

As shown in Figure 7, the adsorption capacity of Pb(II) on Na_2_TiSiO_5_ has a continuous growth with the increase in pH values in the range of 4–7. The modification of pH influences the Pb(II) ions distribution and the adsorbent site’s protonation/deprotonation on the adsorbent’s surface. At the acidic pH, the electrostatic repulsion prevents the effective adsorption of ions due to the protonated surface of the Na_2_TiSiO_5_ adsorbent and positive charge of Pb(II) ions [33,38]. As the pH gradually attains 7.0, the Na_2_TiSiO_5_ adsorbent tends to have negative charges on the surface, due to the superficial deprotonation, which leads to more accessible active sites on the Na_2_TiSiO_5_ particles’ surface. Therefore, the electrostatic attraction of Pb(II) ions and negatively-charged superficial active sites leads to a higher and more convenient adsorption and uptake of Pb(II) ions [38].

The kinetics and dynamics of adsorption of Pb(II) on Na_2_TiSiO_5_ have been investigated with the pseudo first-order (PF-order) kinetics model expressed as Equation (1) [39,40], the pseudo second-order (PS-order) kinetics model expressed as Equation (2) [37,40], as well as the intraparticle diffusion model (IPD) expressed as Equation (3) [39,40].
(1)$$\log\left(q_{e}-q_{t}\right)=\log\left(q_{e}\right)-\frac{k_{1}t}{2.303}\;$$
(2)$$\frac{t}{q_{t}}=\frac{1}{k_{2}q_{t}^{2}}+\frac{1}{q_{e}}t,$$
(3)$$q_{t}=k_{i}t^{\frac{1}{2}}+I$$
where *k_1_* represents the PF-order rate constant (L/min), *k_2_* represents the rate constant of PS-order adsorption (g/mg.min), *k_i_* represents the IPD rate constant (mg/g min^0.5^), and *I* represent the intercept. The kinetics parameters (*q_1_*, *k_1_*, *q_2_*, *k_2_*, *k_i_*, *I*) have been calculated depending on the plot of *log*(*q_e_ − q_t_*) vs. *t*, *t/q_t_* vs. *t*, and *q_t_* vs. *t^1/2^* [1,14].

The Elovich model is widely used to describe the kinetics of chemisorptions and may be expressed as Equation (4) [34,36]:(4)$$q_{t}=\alpha +\beta \;lnt$$
where *α* represents the rate of chemisorption at zero coverage (mg/g min) and *β* represents the desorption rate constant (g/mg). These constants have been calculated having in view the slope and the intercept of the *q_t_* vs. *lnt* plots.

Bangham model is needed to check if pore diffusion is the only rate-controlling step in the adsorption system using the kinetic data [49]. This model can be expressed as Equation (5).
(5)$$log\left\lfloor log\left(\frac{C_{0}}{C_{0}-q_{t\;}m}\right)\right\rfloor =log\left(\frac{k_{0}m}{2.303\;V}\right)+\alpha \;log\left(t\right)$$

Kołodyńska et al. [40] considers that dynamic behavior of the system and the sorption process rate are considered the most important factors for the adsorption process design. The sorption process in the aqueous media takes place in the following steps: (1) The transport of the adsorbate from the bulk phase to the adsorbent exterior surface; (2) The transport of the adsorbate into the adsorbent by pore diffusion and/or surface diffusion (intraparticle diffusion); (3) The ions adsorption on the adsorbent’s surface. The slowest of these three steps establishes the overall rate of the sorption process. It can be appreciated that there is a strong dependence between the sorption kinetics and the chemical and/or physical characteristics of the adsorbed material fact that influences the sorption mechanism. The kinetic parameters, the rate constants, the equilibrium sorption capacities, and the related correlation coefficients, for the five kinetic models are presented in Table 1.

Figure 8, Figure 9 and Figure 10 present the plots of the linearized form of the PF-order model, PS-order model, IPD model, Elovich model, and Bangham model for all studied initial ion concentrations. To calculate the kinetics parameters (*q_1_*, *k_1_*, *q_2_*, *k_2_*, *k_i_*, and *I*), the slopes and intercepts of the plot of *log(q_e_ − q)* vs. *t*, *t/q_t_* vs. *t*, and *q_t_* vs. *t^1/2^* have been used.

By comparing and analyzing the data from Table 1 and the plots in Figure 8, Figure 9, Figure 10, Figure 11 and Figure 12, it can be stated that the adsorption kinetics of Pb(II) on Na_2_TiSiO_5_ are better described with the PS-order kinetic model than the other kinetic models [34,36]. The biggest fitting correlation coefficients was obtained for the PS-order kinetic model (R^2^: 0.996–0.999) comparing to the values obtained for the other models: PF-order (R^2^: 0.962–0.989), the interparticle diffusion model (R^2^: 0.839–0.906), the Elovich model (R^2^: 0.942–0.960), and Bangham model (R^2^: 0.877–0.944). The regression coefficient (R^2^) reflects the agreement between experimental results and model predictions. Therefore, it could be stated that the adsorption process was mainly the chemisorption process [33].

The kinetic data indicates that the mechanism of Pb(II) adsorption by the Na_2_TiSiO_5_ is complex and supposedly is an association of external mass transfer, intraparticle diffusion onto the micropores of Na_2_TiSiO_5_ and sorption processes. Furthermore, it can be said that if the kinetic process fits in the PS-order model, the chemisorption process is assumed [9,34]. When there is a limited rate of chemisorption, the inner-sphere complexation and precipitation involves the Pb(II) ions sorption and the role of electrostatic ion exchange can be neglected [34,35,36]. 

### 2.3. Adsorption Isotherms

Adsorption isotherms have an important role in optimizing experimental design and emphasize the interrelationships between adsorbates and adsorbents [4,9,34,36]. To determine the maximum adsorption capacity of Na_2_TiSiO_5_, the data obtained at the time of equilibrium have been modelled using isotherm equations. 

Langmuir, Freundlich, Halsey, Temkin, Redlich–Paterson, and Dubinin–Radushkevich isotherm models have been used to describe the adsorption of Pb(II) onto Na_2_TiSiO_5_ [37,38,39,40]. 

The non-linearized form of the Langmuir isotherm model is expressed as Equation (6) [40]:(6)$$q_{e}=\frac{q_{0}K_{L}C_{e}}{1+K_{L}C_{e}}$$
where *q_e_* represents the adsorption capacity at equilibrium (mg/g), *C_e_* represents the equilibrium concentration of metal ion (mg/L), *q_0_* (mg/g), *K_L_* represents the characteristic of Langmuir equation (L/mg) and have been determined from its linearized form (plots of 1/*q_e_* vs. 1/*C_e_*). 

The non-linearized form of Freundlich isotherm model expressed as equation (Equation (7)):(7)$$q_{e}=K_{F}C_{e}^{\frac{1}{n}}$$
where *K_F_* represents Freundlich adsorption capacity (mg/g) and *1/n* represents the Freundlich constant related to the surface heterogeneity. Equation (7) should be linearized to calculate *K_F_* and *n* (plots of *log q_e_* vs. *log C*_e_). 

The Halsey model, expressed as Equation (8), is applied to a multilayer adsorption system with distance from the surface [33,34,35,36,37,38,39,40]:(8)$${q_{e}=\left(\frac{K_{H}}{C_{e}}\right)}^{1/n_{H}}$$
where *q_e_* represents the adsorption capacity at equilibrium (mg/g) and *C_e_* represents the equilibrium concentration of Pb(II) (mg/L). *K_H_* and *n_H_* represent the characteristic of the Halsey equation and can be obtained from its linearized form (plots of *log q_e_* vs. *log C_e_*). 

The Redlich–Paterson model expressed as Equation (9) can be used in either heterogeneous or homogeneous systems [40]:(9)$$q_{e}=\frac{K_{RP}C_{e}}{1+a_{R}C_{e}^{\beta }}$$
where *K_RP_*, *a_R_*, and *β* are Redlich–Paterson constants. The Equation (9) can be linearized to calculate the parameters *K_RP_* and *a_R_* for values of *β* between zero and 1 (plots of 1/*q_e_* vs. 1/*C_e_*). The Redlich–Paterson is a combination of Langmuir and Freundlich models and is the most used three-parameter isotherm model.

The Temkin isotherm, expressed as Equation (10), does not take into account the extremely low and high values of the concentrations, containing a factor that represents the interactions between adsorbent and adsorbate [40,41].
(10)$$q_{e}=\frac{RT}{b}ln\left(K_{T}C_{e}\right)$$
where *T* represents the absolute temperature (K), *R* represents the universal gas constant (8.314 J/mol K), *K_T_* represents the equilibrium binding constant (L/mg), and *b* represents Temkin constant or the maximum binding energy (kJ/mol) [40].

Dubinin–Kaganer–Radushkevich isotherm, expressed by Equation (11), is applied to illustrate the adsorption mechanism with a Gaussian energy distribution onto a heterogeneous surface and is mostly used to distinguish between physiosorption and chemisorption process [40,49].
(11)$$ln\left(q_{e}\right)=ln\left(q_{s}\right)-k_{ad}\varepsilon^{2}$$
where *q_s_* represents theoretical isotherm saturation capacity (mg/g); *q_e_* represents amount of adsorbate in the adsorbent at equilibrium (mg/g); *K_ad_* represents the Dubinin–Kaganer–Radushkevich isotherm constant (mol^2^/kJ^2^) and *ε* represents the Dubinin–Radushkevich isotherm constant. The parameter *ε* can be calculated using Equation (12) [40]:(12)$$\varepsilon =RT\;ln\left(1+\frac{1}{C_{e}}\right)$$
where *R*, *T*, and *C_e_* are the gas constant (8.314 J/mol K), absolute temperature (K), and adsorbate equilibrium concentration (mg/L). 

The isotherm parameters and correlation coefficients have been obtained from the intercepts and slopes of the respective plots (Table 2). Figure 13 presents the adsorption isotherms obtained for Pb(II) adsorption on Na_2_TiSiO_5_ at 293, 303, 313 K temperatures.

The quantitative monolayer pollutant adsorption on the adsorbent surface has been defined using the Langmuir equation. According to the Langmuir model, all the active sites on adsorbent surface have the same adsorption energy. Contrastingly, the Freundlich isotherm model is suitable for multilayer adsorption over the heterogeneous adsorbent surface [40].

As shown in Figure 13, all models well fit the equilibrium data with coefficients (R^2^) higher than 0.90. However, Langmuir and Redlich–Paterson models were able to provide a better fit (R^2^: 0.9936–0.9996) to the experimental data than Freundlich and Halsey (R^2^ 0.9389–0.9912), Temkin (R^2^ 0.9835–0.9928), and Dubinin–Kaganer–Radushkevich model (R^2^ 0.9068–0.9928). It indicates that the Pb(II) adsorption onto Na_2_TiSiO_5_ is a major monolayer adsorption process. The maximum adsorption capacity that has been achieved at 298 K using Na_2_TiSiO_5_, based on Langmuir isotherm, was 155.71 mg/g.

The removal efficiency of the synthesized Na_2_TiSiO_5_ is comparable to or even better than other the materials previously investigated as Pb(II) sorbents (Table 3).

### 2.4. Thermodynamics of Pb(II) Adsorption

To conclude whether the adsorption process is spontaneous or not it is necessary to consider the process’s thermodynamic characteristics. Gibb’s free energy change (*ΔG*°) is the fundamental criterion of reaction’s spontaneity; if *ΔG*° has a negative value at a given temperature then the reaction take place spontaneously [12,48]. The thermodynamic parameters for the adsorption process (Gibb’s free energy change (*ΔG*°), enthalpy change (*ΔH*°), and entropy change (*ΔS*°)) is given by the Equations (13) and (14) [48,49,50]:(13)$$\Delta G{}^{\circ}=-RT\ln K_{L}$$
(14)ΔG°=ΔH°−TΔS°
where *K_L_* has been obtained from the Langmuir equation, *T* represent the absolute temperature in K, and *R* represent the universal gas constant (R = 8.314 J/mol K).

The thermodynamic parameters for the adsorption of Pb(II) ions on the Na_2_TiSiO_5_ at various temperatures have been calculated using Equations (13) and (14) and summarized in Table 4. 

The plot of Gibb’s free energy change (*ΔG°*) vs. temperature; *T* is presented in Figure 14.

The positive values of enthalpy change (*ΔH°*) denote that the studied adsorption processes are endothermic. Moreover, the negative values of Gibb’s free energy change (*ΔG*°) indicate the spontaneous behavior of the adsorption processes [1]. The decrease in the value of *ΔG*° with the increase in temperature point out that the reaction was more spontaneous at high temperatures; the adsorption process is favored when temperature increases [49,50]. The positive values of entropy change (*ΔS*°) reflect the affinity of the Na_2_TiSiO_5_ for Pb(II) ions, thus it can be interpreted that some structural changes at the solid–liquid interface also take place [1,6,46].

## 3. Materials and Methods

### 3.1. Synthesis and Characterization

#### 3.1.1. Materials

Sodium titanosilicate powders Na_2_TiSiO_5_ have been obtained using the modified sol–gel method, starting from NaOH (5 M), titanium tetraisopropoxide Ti(O_3_C_3_H_7_)_4_ (TEOS), and tetraethylorthosilicate Si(OCH_2_CH_3_)_4_ (Merck KGaA, Darmstadt, Germany; reagents with purities > 99%). The precursor has been obtained from starting materials mixed in ideal cation stoichiometry for Na_2_TiSiO_5_. After stirring at 80 °C and drying at 110 °C, the precursor has been heat-treated between 500–900 °C for two hours.

#### 3.1.2. Characterization of the Prepared Sodium Titanosilicate Powders

The obtained product, Na_2_TiSiO_5_ has been characterized using X-ray diffraction (XRD), differential thermal analysis (TG-DTG), scanning electron microscopy coupled with energy dispersive X-ray spectroscopy (EDX), and high-resolution electron microscopy (HRTEM). The thermal decomposition behavior of the gel precursor of has been examined by means of thermogravimetry (TG-DTG) with a MOM type C derivatograph. XRD analysis has been performed using a BRUKER D8 X-ray diffractometer with CuK_α_ radiation beam (λ = 0.154060 nm) and with a step size of 0.05° and a resolution of 0.01. The samples of Na_2_TiSiO_5_ have been packed into a flat aluminum sample holder, and the X-rays have been generated at 30 kV and 30 mA. Scans have been carried out at 2 min^−1^ for 2θ values between 25 and 50°.

Fourier transform infrared absorption (FTIR) spectra has been performed using a BRUKER VECTOR 22 spectrometer on samples embedded in KBr pellets.

Electron microscopy has been used to evaluate the morphology and microstructures of Na_2_TiSiO_5_ nanopowder. SEM and EDX have been recorded using a JEOL JMS 5800L electron microscope. To obtain the SEM imagines the samples have been coated with gold and examined in the as-fired condition, i.e., without polishing. The bright field–transmission electron microscopy (BF-TEM) investigations have been performed using a Philips CM 120 ST electron microscope, which at 100 kV provides a resolution of 2 Å [32]. TEM samples have been prepared by dispersing fine powder grinded from bulk sample in ethanol, followed by ultrasonic agitation, and then deposed onto a carbon-enhanced copper grid [35]. The distribution of nanoparticles size has also been studied.

### 3.2. Adsorption Capacity Studies

The adsorption of Pb(II) by Na_2_TiSiO_5_ nanopowder has been studied using a batch technique, including effects of contact time, pH, temperature, and initial metal ion concentration. In all experiments distilled water was used. Lead stock solutions (0.1 M) have been prepared from lead nitrate (Pb(NO_3_)_2_ (Merck reagent). Batch adsorption experiments were performed at 298 K, 303 K, and 313 K temperatures. A total of 50 mL of Pb(II) solutions with the initial concentration from 1 × 10^−2^ M to 7 × 10^−2^ M were placed in flasks with 0.5 g Na_2_TiSiO_5_ and shaken at a constant speed (200 rpm) for different contact times (5–360 min). The water/solid ratio was 100. The samples were magnetic stirred using a laboratory stirrer at different time intervals. After that, the suspensions was centrifuged for 5 min at 4000 rpm and the supernatants were collected and analyzed for Pb(II) concentration. The lead ions concentration, before and after equilibrium, were determined using a flame atomic adsorption spectrophotometer (FAAS) and ZEENIT 700 Atomic Absorption Spectrometer. The determination of the Pb(II) concentration was carried out at a wavelength of 283.3 nm using a monoelement lamp for the concentration range 0.01–4 mg/L, with a correlation coefficient of the calibration curve of 0.9989. The experiments have been performed in duplicate or until the average values were obtained (±0.005–0.01 mg/g for the amount adsorbed). The pH effect on Pb(II) removal efficiency was investigated at a pH range of 1.0–7.0 at room temperature. A digital 720 Inolab Multiparameter was used for pH measuring.

Kinetic studies of Pb(II) removal were performed at different initial concentrations of the Pb(II) solution (1·× 10^−2^ M, 3·× 10^−2^ M, 5·× 10^−2^ M and 7·× 10^−2^ M) where in the extent of adsorption was investigated as a function of time. The effect of temperature on the kinetics of Pb(II) removal has been studied at 298 K, 303 K, and 313 K and as the effect of contact time. 

The amounts of Pb(II) removal by the Na_2_TiSiO_5_ adsorbents has been calculated using Equation (15):(15)$$q_{t}=\frac{\left(C_{0}-C_{t}\right)V}{m}$$
where

*q_t_* = the adsorption amount of Pb(II) at time *t* (mg/g);*m* = the weight of Na_2_TiSiO_5_ sample (g);*V* = the total volume of solution (L);*C_0_* = the initial concentrations of Pb(II) ions in solution (mol/L);*C_t_* = the equilibrium concentrations of Pb(II) ions in solution at time *t* (mol/L).

The removal efficiency is calculated using Equation (16):(16)$$E_{f}=\frac{\left(C_{0}-C_{t}\right)}{C_{0}}\times 100$$
where *t* represents the equilibrium contact time, *C_t_* is equal to *C_e_*, *q_t_* is equal to *q_e_*, and the removal amount of Pb(II) at equilibrium, *q_e_*, has been calculated in accordance with Equation (15).

## 4. Conclusions

This paper represents an original contribution on Pb(II) removal from aqueous media using a new adsorbent, nanopowder sodium titanosilicate, Na_2_TiSiO_5_. The new adsorbent Na_2_TiSiO_5_, has been successfully prepared by sol-gel method. The structural and morphological characterization of synthesized product has been made using thermal analyses (TG-DTG), X-ray diffraction (XRD), and electron microscopy (SEM and TEM) analysis. The synthesized Na_2_TiSiO_5_ nanoparticles have been placed in the size range of 3–16 nm with 7.45 nm mean diameter and proved efficiency and high sensitivity for uptake the Pb(II) ions from aqueous media. 

Adsorption properties of the obtained Na_2_TiSiO_5_ nanopowder has been investigated for Pb(II) removal from aqueous media in different experimental conditions such as the contact time, the initial metal concentration, pH, and temperature. The Pb(II) adsorption on Na_2_TiSiO_5_ has been discussed according to the isotherms, kinetics, and thermodynamics models. Five kinetic models including the PF-order equation, PS-order equation, intraparticle diffusion equation, Elovich equation, and Bangham’s equation have been selected to follow the adsorption process of the Pb(II) ions on Na_2_TiSiO_5_. 

The adsorption kinetics of Pb(II) have been better described using the PS-order kinetic model with the biggest fitting correlation coefficients (R^2^: 0.996–0.999). The kinetic data indicate that the mechanism of Pb(II) adsorption by the Na_2_TiSiO_5_ powder is a complex one and probably is a combination of external mass transfer, intraparticle diffusion through the micropores of Na_2_TiSiO_5_, and the sorption processes. 

The Pb(II) adsorption isotherms have been also modelled using Langmuir, Freundlich, Halsey, Temkin, Redlich–Paterson, and Dubinin–Kaganer–Radushkevich isotherm models. By comparing the values of linear regression coefficient (R^2^) of the examined six isotherm models, it can be concluded that the Langmuir and Redlich–Paterson isotherm models, gave much better fitting than the other isotherm models (R^2^: 0.9936–0.9996). Consequently, the adsorption behavior of Pb(II) ions on Na_2_TiSiO_5_ nanopowder can be well described using these two isotherm models. The results revealed a maximum adsorption capacity of 155.71 mg/g at 298 K and a very high adsorption rate at the beginning, more than 85% of the total amount of Pb(II) being removed within the first 120 min, depending on the initial concentration.

The calculated thermodynamic parameters indicate that the Pb(II) adsorption is an endothermic process, with increased entropy and a spontaneous reaction. Certainly, further research should be carried out in this direction, especially in terms of the assessment of the stability and reusability of Na_2_TiSiO_5_ for multi-component heavy metals sorption.

## Figures and Tables

**Figure 1 ijms-24-13789-f001:**
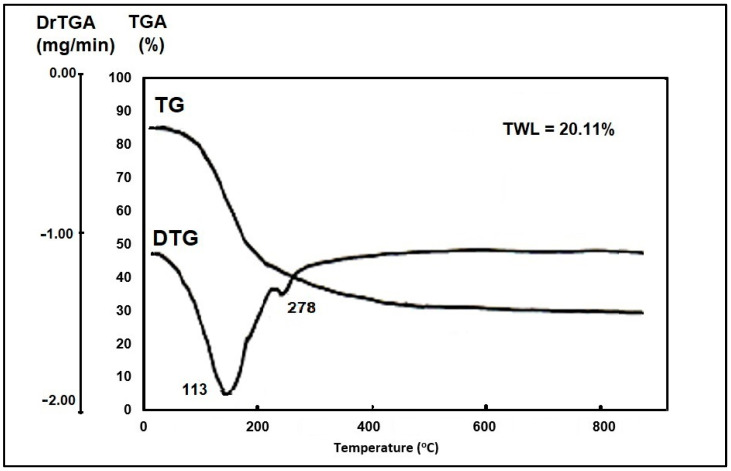
TG–DTG curves of the synthesized Na_2_TiSiO_5_.

**Figure 2 ijms-24-13789-f002:**
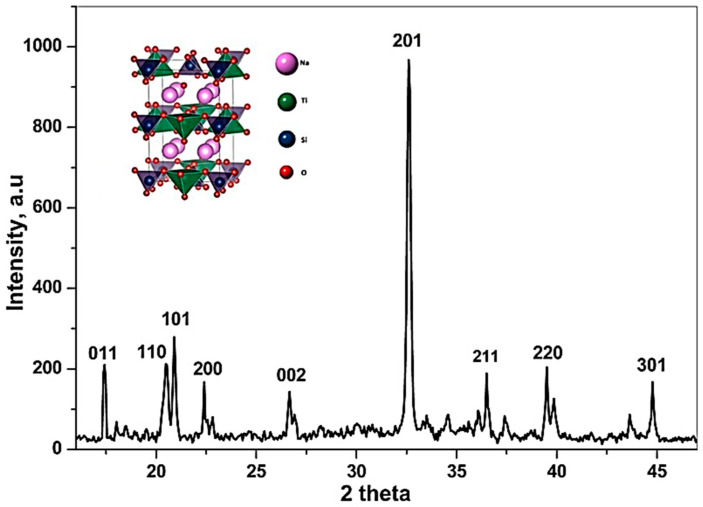
Indexed XRD patterns of nanocrystalline Na_2_TiSiO_5_ powders.

**Figure 3 ijms-24-13789-f003:**
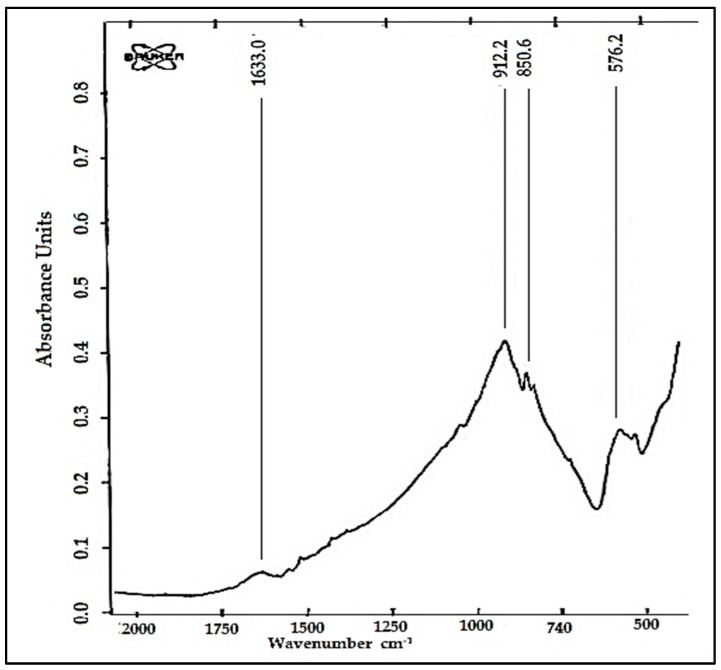
FTIR spectra of Na_2_TiSiO_5_ powders.

**Figure 4 ijms-24-13789-f004:**
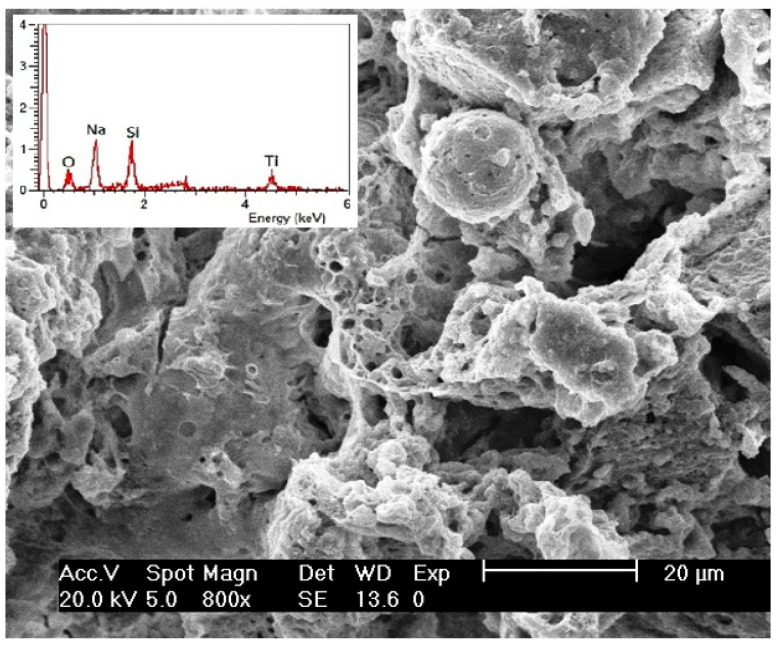
SEM micrograph of Na_2_TiSiO_5_ powders with EDX spectra.

**Figure 5 ijms-24-13789-f005:**
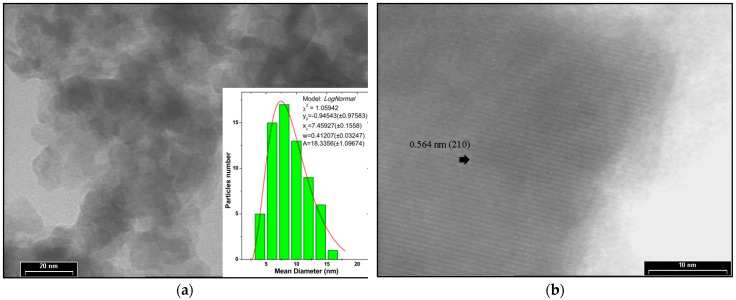
(**a**) BFTEM image of Na_2_TiSiO_5_ powder with insertion of Feret histogram of diameter; (**b**) HRTEM imagine of Na_2_TiSiO_5_ powders.

**Figure 6 ijms-24-13789-f006:**
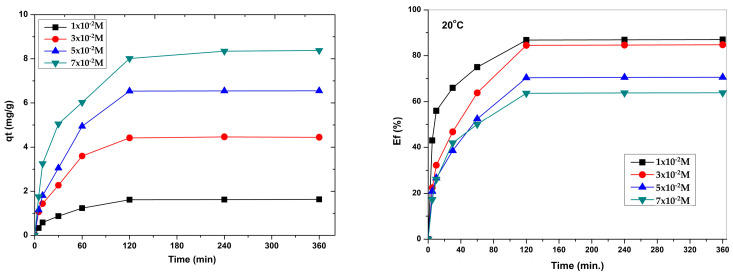
The evolution of the adsorption capacity (**a**) and removal efficiency of Pb(II) uptake (**b**) on the Na_2_TiSiO_5_ particles.

**Figure 7 ijms-24-13789-f007:**
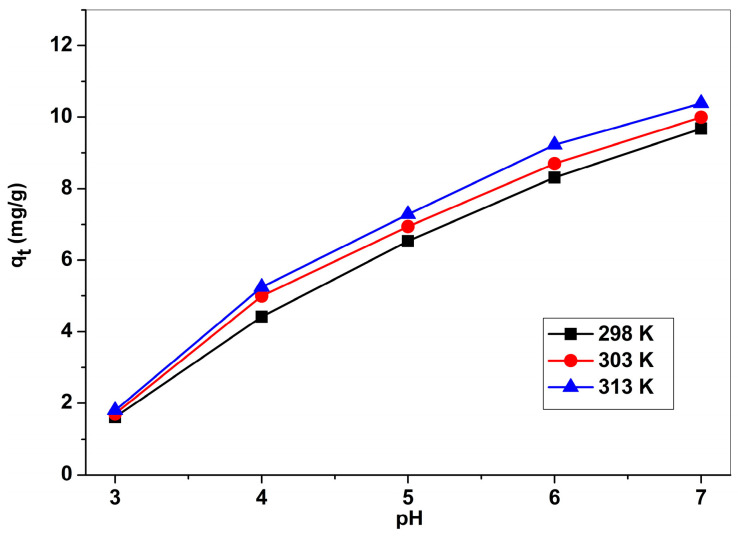
The effect of pH on the adsorption capacity of Pb(II) on the surface of Na_2_TiSiO_5_.

**Figure 8 ijms-24-13789-f008:**
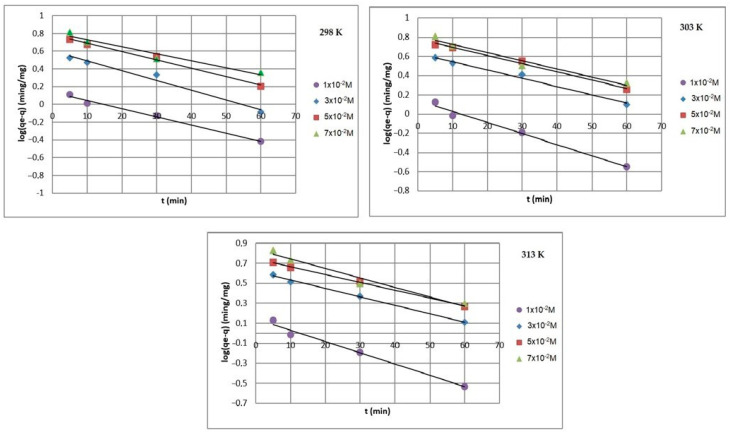
PF-order plots of Pb(II) ions adsorption on Na_2_TiSiO_5_.

**Figure 9 ijms-24-13789-f009:**
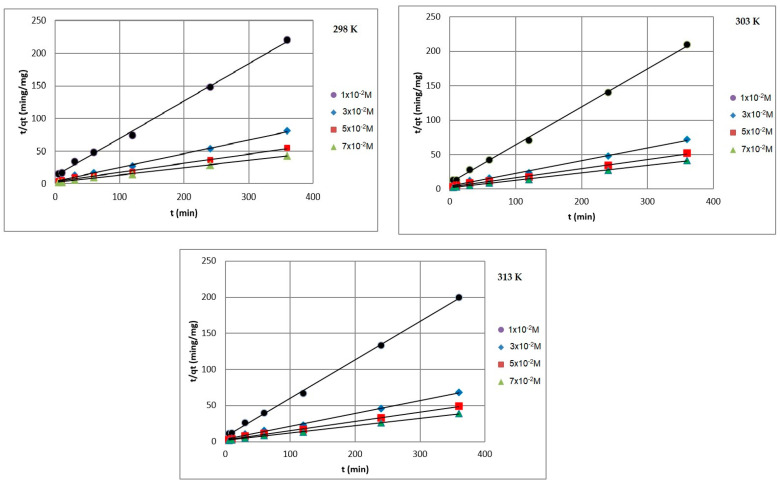
PS-order plots of Pb(II) ions adsorption on Na_2_TiSiO_5_.

**Figure 10 ijms-24-13789-f010:**
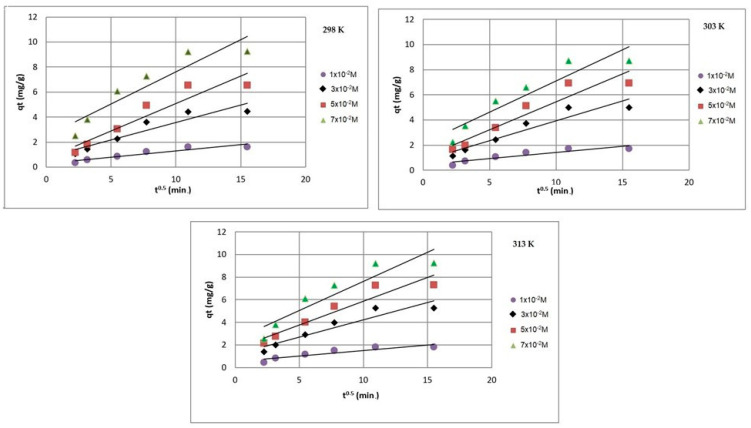
Intraparticle diffusion model plots of Pb(II) ions adsorption on Na_2_TiSiO_5_.

**Figure 11 ijms-24-13789-f011:**
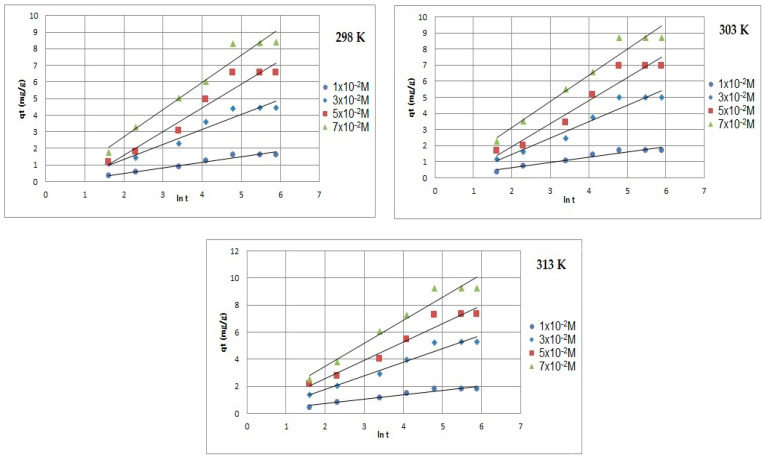
Elovich model plots of Pb(II) ions adsorption on Na_2_TiSiO_5_.

**Figure 12 ijms-24-13789-f012:**
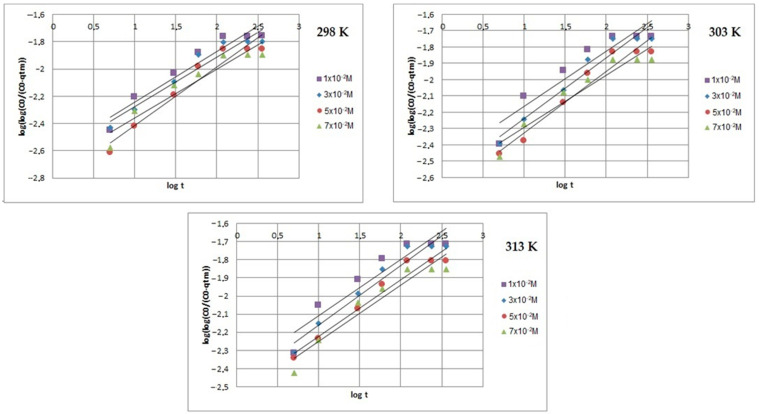
Bangham’s model plots of Pb(II) ions adsorption on Na_2_TiSiO_5_.

**Figure 13 ijms-24-13789-f013:**
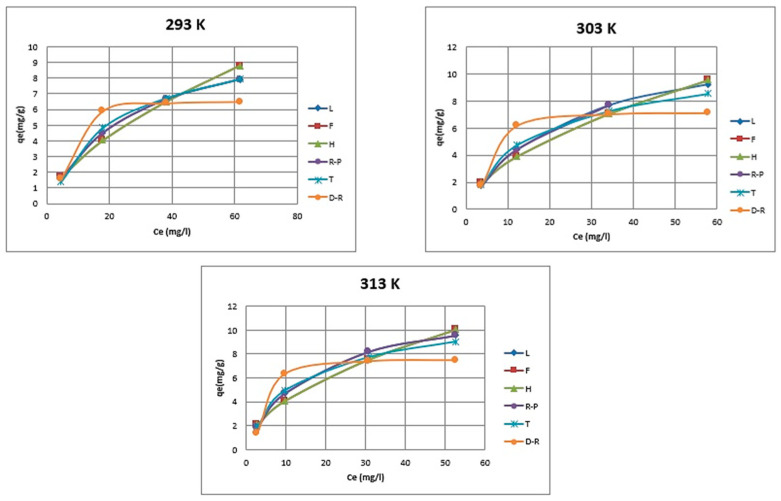
Adsorption Isotherms for different temperatures. (L—Langmuir isotherm, F—Freundlich isotherm, H—Halsey, R-P—Redlich–Paterson, T—Temkin, D-R—Dubinin—Kaganer–Radushkevich).

**Figure 14 ijms-24-13789-f014:**
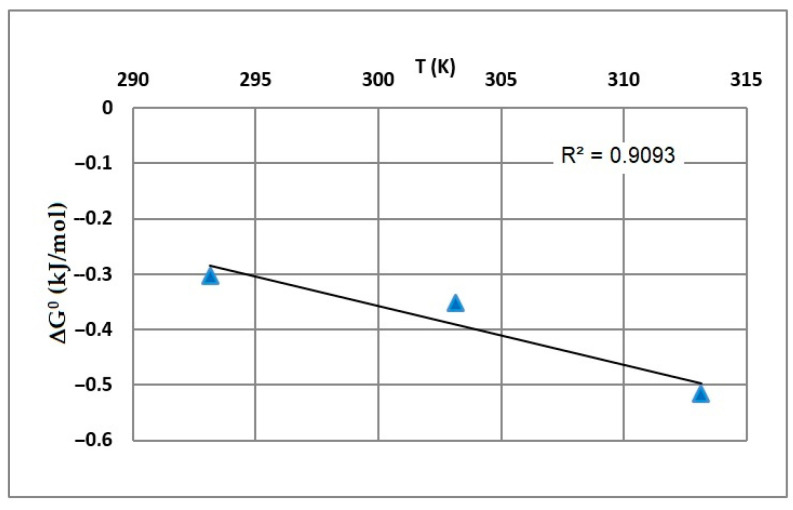
Plot of Gibbs free energy change (*ΔG*°) vs. temperature, *T* (*K*).

**Table 1 ijms-24-13789-t001:** Kinetic parameters Pb(II) ions adsorption on Na_2_TiSiO_5_ at different temperatures.

		PF-Order Model				PS-Order Model				Intraparticle Diffusion Model (IPD)				
	q_e,exp_ (mg/g)	q_1_ (mg/g)	k_1_ (1/min)	R^2^	Chi Square χ^2^	q_m_ (mg/g)	k_2_ * 100 (g/mg.min)	R^2^	Chi Square χ^2^	q_m_ (mg/g)	k_i_ (mg/g.min^0.5^)	I	R^2^	Chi Square χ^2^
293 K														
1 × 10^−2^ M	1.618	1.487	0.021	0.989	0.482	1.485	3.097	0.998	0.038	1.392	0.101	0.290	0.886	0.155
3 × 10^−2^ M	4.419	4.207	0.025	0.978	0.769	4.069	1.173	0.997	0.166	3.835	0.278	0.789	0.879	0.431
5 × 10^−2^ M	6.530	6.046	0.022	0.987	0.584	5.835	0.609	0.996	0.220	5.521	0.440	0.697	0.890	0.686
7 × 10^−2^ M	8.314	7.377	0.018	0.944	4.737	7.630	0.629	0.998	0.174	7.170	0.498	1.716	0.882	0.835
303 K														
1 × 10^−2^ M	1.708	1.639	0.027	0.986	0.450	1.601	3.914	0.999	0.029	1.514	0.099	0.432	0.841	0.222
3 × 10^−2^ M	4.995	4.531	0.020	0.988	1.618	4.527	0.933	0.996	0.216	4.251	0.315	0.798	0.888	0.389
5 × 10^−2^ M	6.940	6.276	0.020	0.989	2.131	6.260	0.644	0.996	0.349	5.876	0.444	1.017	0.899	0.544
7 × 10^−2^ M	8.698	7.890	0.020	0.964	5.683	8.089	1.095	0.998	0.138	7.586	0.495	2.164	0.877	0.764
313 K														
1 × 10^−2^ M	1.796	1.717	0.026	0.984	0.737	1.690	4.176	0.999	0.027	1.597	0.099	0.516	0.839	0.207
3 × 10^−2^ M	5.248	4.733	0.019	0.988	3.160	4.837	1.075	0.998	0.183	4.522	0.308	1.147	0.898	0.359
5 × 10^−2^ M	7.280	6.462	0.018	0.989	6.028	6.715	0.781	0.997	0.389	6.266	0.420	1.671	0.906	0.406
7 × 10^−2^ M	9.225	9.225	0.094	0.962	2.021	8.640	0.751	0.999	0.103	8.125	0.515	2.481	0.864	0.870
				**Elovich Model**					**Bangham’s Model**					
		**C_0_** **(mg/L)**	**q_e,exp_** **(mg/g)**	**q_m_ (mg/g)**	**α_E_**	**β**	**R^2^**	**Chi Square χ^2^**	**q_m_ (mg/g)**	**α_B_**	**K_0_** **(L/g)**	**R^2^**	**Chi Square χ^2^**	
293 K														
1 × 10^−2^ M		20.7	1.618	1.4097	−0.1701	0.33	0.958	0.0529	1.3499	0.3723	0.0005578	0.9237	0.19241	
3 × 10^−2^ M		62.1	4.419	3.8697	−0.4538	0.903	0.942	0.2230	3.6950	0.3628	0.0005318	0.933	0.54020	
5 × 10^−2^ M		103.5	6.530	5.5846	−1.2667	1.431	0.947	0.3603	5.2571	0.4286	0.933	0.9337	1.03999	
7 × 10^−2^ M		144.9	8.314	7.2678	−0.5856	1.640	0.960	0.2519	7.0008	0.3553	0.0004463	0.908	0.98176	
303 K														
1 × 10^−2^ M		20.7	1.708	1.5290	−0.0331	0.326	0.947	0.0748	1.4834	0.3365	0.0007287	0.8775	0.24375	
3 × 10^−2^ M		62.1	4.995	4.2929	−0.6037	1.022	0.946	0.2389	4.0837	0.3696	0.0005698	0.9446	0.51874	
5 × 10^−2^ M		103.5	6.940	5.9410	−0.9362	1.436	0.944	0.4097	5.6319	0.3757	0.0004566	0.9439	0.75381	
7 × 10^−2^ M		144.9	8.698	7.6723	−0.1217	1.628	0.960	0.2255	7.4340	0.3188	0.0005649	0.9214	0.80227	
313 K														
1 × 10^−2^ M		20.7	1.796	1.6122	0.0501	0.326	0.946	0.0717	1.6669	0.346	0.0007843	0.8781	0.27829	
3 × 10^−2^ M		62.1	5.248	4.5719	−0.2361	1.004	0.955	0.1586	4.4007	0.3271	0.0007536	0.9442	0.41166	
5 × 10^−2^ M		103.5	7.280	6.3344	−0.1876	1.362	0.951	0.2472	6.0978	0.3103	0.0006770	0.9531	0.47265	
7 × 10^−2^ M		144.9	9.225	8.2086	0.0814	1.697	0.957	0.2527	7.9746	0.3087	0.0006366	0.9166	0.87329	

**Table 2 ijms-24-13789-t002:** Isotherm parameters and correlation coefficients for Pb(II) sorption on the Na_2_TiSiO_5_.

Model		293 K	303 K	313 K
Langmuir	q_0_ (mg/g)	11.4155	13.0039	12.4688
	K_L_ (L/mg)	0.0365	0.0423	0.0621
	χ^2^	0.9990	0.9808	0.9797
	R^2^	0.9996	0.9933	0.9936
Freundlich	K_F_ [(mg/g)(L/mg)^1/n^]	0.6549	0.9363	1.2103
	n	1.5886	1.7476	1.8744
	χ^2^	0.9957	0.9488	0.9415
	R^2^	0.9912	0.9439	0.9389
Halsey	K_H_	1.9587	1.1220	0.6992
	n_H_	−1.5886	−1.7476	−1.8744
	χ^2^	0.9957	0.9488	0.9415
	R^2^	0.9912	0.9439	0.9389
Redlich–Paterson	K_RP_ (L/g)	0.4163	0.5497	0.7744
	a_RP_ (L/mg)	0.0365	0.0423	0.0621
	β	1.0000	1.0000	1.0000
	χ^2^	0.9990	0.9808	0.9797
	R^2^	0.9996	0.9933	0.9936
Temkin	b	973.1868	999.528	1013.662
	K_T_ (L/mg)	0.384643	0.575466	0.805618
	χ^2^	0.9923	0.9984	0.9964
	R^2^	0.9835	0.9928	0.9885
Dubinin–Kaganer–Radushkevich	K_ad_ (mol^2^/kJ^2^)	0.000006	0.000004	0.000003
	q_s_ (mg/g)	6.5476	7.1721	7.5217
	χ^2^	0.8261	0.9034	0.8807
	R^2^	0.9068	0.9460	0.9425

**Table 3 ijms-24-13789-t003:** The maximum adsorption capacity of Pb(II) sorption on other materials.

Adsorbent	Adsorption Capacity (mg/g)	Reference
CMA (porous silica)	196.35	[8]
MIL-88A-LDHs	512.8	[11]
Modified biochar	145.0	[33]
Fe_3_O_4_@SBA-15-Gd	175.24	[38]
NiFe_2_O_4_/MnO_2_	85.78	[42]
MPH-220 (carbonaceous)	174.75	[43]
Fe_3_O_4_@BC/APTES	64.92	[44]
CuO	3.31	[45]
Graphene-ZnO	23.42	[46]
Co_3_O_4_ co-doped TiO_2_	114.05	[47]
Na_2_TiSiO_5_	155.71	This work

**Table 4 ijms-24-13789-t004:** Adsorption thermodynamic parameters of Pb(II) ions on Na_2_TiSiO_5_.

*T* (K)	*ΔG*° (kJ/mol)	*ΔS*° (J/mol K)	*ΔH*° (kJ/mol)	R^2^
293	−0.30317	10.7	2.8412	0.9093
303	−0.35144			
313	−0.51637			

## Data Availability

Not applicable.

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
