# Peer review of "New Approaches for Pb(II) Removal from Aqueous Media Using Nanopowder Sodium Titanosilicate: Kinetics Study and Thermodynamic Behavior"

_ijms, 2023, doi:10.3390/ijms241813789_

Round 1

Reviewer 1 Report

The manuscript ``New approaches for Pb (II) removal from aqueous media by nanopowder sodium titanosilicate: Kinetics study and thermodynamic behavior`` by Ionela Carazeanu Popovici et al. is an interesting work. The authors put a clear effort into the study, which is supported by many experimental findings. However, it still contains significant technical and scientific problems. In my opinion, the present work should be improved before being considered for publication.

-          Please check the written references number in the manuscript. For example in introduction missing references between 6 and 17. Lines 48,50. Also, check if there was a good comparison of results with literature.

-          Line 98, typing mistake, ``The linearized Langmuir and Freundlich equations``. In the calculation you used a non-linear form of the equations.

-          The manuscript could be better organized. For easier reading and interpretation of the manuscript, the section Material and Methods should come before the Results and Discussion.

-          The characterization of the prepared sodium titanosilicate powders is well done. In aim to determine the changes in the chemical structure, do you have any characterization results of the prepared material after Pb (II) adsorption?

-          Lines 109-110, ``The interpretation is supported by appearance of endothermic peak in DTG curve at 113°C corresponding to water adsorbed in the pores and at the 178°C corresponding to desorption of structural water``. Please, if possible, support the sentence with literature.

-          Also lines 135-137, support the sentences with literature.

-          In Figure 6. Could you put q (mg/g) with Ef (%) on the y axis as a dependent variable? The adsorption capacity was used in all equation and discussion so in my opinion Ef (%) is less precious and less important.

-          Please check the written equations. For example Eq 2, line 193 instead of a minus sign there should be a plus sign. Could you considered using non linear kinetic PFO and PSO forms? The authors are encouraged to see the paper, which maybe could help https://doi.org/10.3390/pr11051327.

-          Line 222, typing mistake, the PF-order was written twice.

-          Figure 7. since it is a log (qe-q), there are no units of measurement.

-          Table 2 please add units of isotherm parameters and Chi-square χ2.

-          In Results and disscusion part, Adsorption isotherms, maybe the author should compare their results clearly with other, similar reported works. Maybe add a comparison in the form of a table or just add it in text.

-          In Abstract, line 20, author mentioned that the maximum capacity was 155.71 mg/g,  in the Results and Discussion part, Adsorption isotherms, that information is nowhere to be seen. Please, improve discussion in this part.

-          Lines 341, 343, 345 are missing numbers of reference.

-          Please, improve Conclusion part. What results did you get? What is novelty and what is the scientific contribution of your work?

The manuscript ``New approaches for Pb (II) removal from aqueous media by nanopowder sodium titanosilicate: Kinetics study and thermodynamic behavior`` by Ionela Carazeanu Popovici et al. is an interesting work. The authors put a clear effort into the study, which is supported by many experimental findings. However, it still contains significant technical and scientific problems. In my opinion, the present work should be improved before being considered for publication.

-          Please check the written references number in the manuscript. For example in introduction missing references between 6 and 17. Lines 48,50. Also, check if there was a good comparison of results with literature.

-          Line 98, typing mistake, ``The linearized Langmuir and Freundlich equations``. In the calculation you used a non-linear form of the equations.

-          The manuscript could be better organized. For easier reading and interpretation of the manuscript, the section Material and Methods should come before the Results and Discussion.

-          The characterization of the prepared sodium titanosilicate powders is well done. In aim to determine the changes in the chemical structure, do you have any characterization results of the prepared material after Pb (II) adsorption?

-          Lines 109-110, ``The interpretation is supported by appearance of endothermic peak in DTG curve at 113°C corresponding to water adsorbed in the pores and at the 178°C corresponding to desorption of structural water``. Please, if possible, support the sentence with literature.

-          Also lines 135-137, support the sentences with literature.

-          In Figure 6. Could you put q (mg/g) with Ef (%) on the y axis as a dependent variable? The adsorption capacity was used in all equation and discussion so in my opinion Ef (%) is less precious and less important.

-          Please check the written equations. For example Eq 2, line 193 instead of a minus sign there should be a plus sign. Could you considered using non linear kinetic PFO and PSO forms? The authors are encouraged to see the paper, which maybe could help https://doi.org/10.3390/pr11051327.

-          Line 222, typing mistake, the PF-order was written twice.

-          Figure 7. since it is a log (qe-q), there are no units of measurement.

-          Table 2 please add units of isotherm parameters and Chi-square χ2.

-          In Results and disscusion part, Adsorption isotherms, maybe the author should compare their results clearly with other, similar reported works. Maybe add a comparison in the form of a table or just add it in text.

-          In Abstract, line 20, author mentioned that the maximum capacity was 155.71 mg/g,  in the Results and Discussion part, Adsorption isotherms, that information is nowhere to be seen. Please, improve discussion in this part.

-          Lines 341, 343, 345 are missing numbers of reference.

-          Please, improve Conclusion part. What results did you get? What is novelty and what is the scientific contribution of your work?

Reviewer 2 Report

Recommendation: Acceptable after major revisions.

The authors have studied Pb (II) removal from aqueous media using sodium titanosilicate nano powder and explained the kinetic and thermodynamics behaviors. This approach could be one of the better approaches for the removal and they have shown the 80-90% of total removal. Although there are extensive studies in this direction still open for studies and interpretations. Nevertheless, I would ask the authors to deeply re-consider their manuscript.

I have minor points mentioned herewith.

1.      There are some typographical errors, like; Na2TiSiO5 could be Na2TiSiO5 and degree Celsius for the temperature representation which should be in superscript.   There are others typographical errors, they should thoroughly consider it.

2.      Figure 1: The axes are missing. They should add that.

3.      Figure 3: Authors could plot Transmittance vs Wavenumber for FTIR spectra.

4.      They should consider more data points for the adsorption behavior, which would be better to fit their models.

5.      I feel that I’m missing more temperature studies instead of three temperatures and detailed thermodynamics behind the adsorption properties.

6.      Authors should rewrite the conclusion part which is just now as take-home approach.  

I feel moderate editing of English language required.

Reviewer 3 Report

I thank the editor for inviting me to review the manuscript.

The authors have reported the “New approaches for Pb (II) removal from aqueous media by nanopowder sodium titanosilicate: Kinetics study and thermodynamic behavior”. The concept is good. It is an interesting article with a clear technological application. On the other hand some improvements have to be performed on the manuscript in order for it to be published.

Comments

1. Keywords should be rewritten

2. L 56-64 should be supported by ref.

3. Authors should discussed the reproducibility of the synthesized material by repeating experiments at least three times.

4. Sections 2.1.1., 2.1.2, and 2.1.3 should supported by references.

5. What is the pH that selected it for kinetics and isotherms? They should study the sorption against pH firstly.

6. Authors should support the discussion by speciation of Pb(II) in the solution at selected pH value.

7. Authors should make comparison of the data with that in the literature.

8. Conclusion should be rewritten

Minor editing of English language required

Round 2

Reviewer 1 Report

I have reviewed the manuscript carefully and there is no need for further revisions.

Reviewer 2 Report

Acceptable in the current form.

Moderate editing of English language  required

Reviewer 3 Report

The manuscript was improved and it can be accepted for publication in present form